# Spatial Identification and Hotspots of Ecological Risk from Heavy Metals in Urban Dust in the City of Cartagena, SE Spain

Pura Marín-Sanleandro, María José Delgado-Iniesta *, Anthony Felipe Sáenz-Segovia and Antonio Sánchez-Navarro

Department of Agricultural Chemistry, Geology and Pedology, Faculty of Chemistry, Campus de Espinardo, University of Murcia, 30100 Murcia, Spain; pumasan@um.es (P.M.-S.); anthonyfelipe.saenzs@um.es (A.F.S.-S.); antsanav@um.es (A.S.-N.)
* Correspondence: delini@um.es; Tel.: +34-868887447

**Abstract:** In the present work, a study has been carried out on the contamination of heavy metals in urban dust deposited on the roads of the city of Cartagena (Spain) in order to know the content of metals such as Ni, Zn, Pb, Cd, Cr and Cu. Likewise, the possible relationship between the concentration of heavy metals and the color of the sample, level of magnetism and traffic density was studied. Contamination was evaluated using several indices such as contamination factor (CF), enrichment factor (EF), geo-accumulation index (Igeo), pollutant load index (PLI) and ecological risk index (RI). A total of 88 samples were taken in the urban area of Cartagena, and the metals were determined by acid digestion and measured by inductively coupled plasma mass spectrometry. The concentration of heavy metals in urban dust from Cartagena was Zn (672 mg kg$^{-1}$) > Cu (248.9 mg kg$^{-1}$) > Pb (227 mg kg$^{-1}$) > Cr (82.7 mg kg$^{-1}$) > Ni (47.7 mg kg$^{-1}$) > Cd (4.1 mg kg$^{-1}$). Contamination levels were high in Pb, Zn, Cd and Cu, according to environmental indices. A correlation was found between magnetism and metal concentration, which was repeated for all metals except Cd. Dark-colored samples contained higher metal concentrations than light-colored samples. Meanwhile, streets with medium and low traffic intensity were found to have higher concentrations of heavy metals. This study's objective was to identify pollution hotspots caused by heavy metals in dust in the urban ecosystem of Cartagena city.

**Keywords:** heavy metals; pollution; urban dust; ecological risk index; traffic intensity; hostspots

## 1. Introduction

Urban dust is contaminated by toxins from a variety of diffuse sources that are difficult to control [1]. Urban dust raises the level of heavy metal exposure of humans and ecosystems in and around urban areas and can have an apparent or hidden impact [2].

The sources of heavy metal pollution can be of anthropogenic or geogenic origin. The main source of pollution is of anthropogenic origin due to activities such as air pollution from vehicles, fossil fuel combustion, agricultural fertilizers, mining, metallurgy, etc. [3,4].

Recently, the World Health Organization has stated that about 7 million people die premature deaths due to cancer and other diseases caused by air pollution [5].

Depending on atmospheric conditions, emitted urban dust particles take more or less time to settle on the surface of roads, parks, roofs of houses or buildings, etc. [6].

Normally, the concentration of heavy metals in dust deposited on roads is higher than that observed in dust from commercial and residential surfaces. On the other hand, these concentrations are higher in highly polluted industrial areas, whereas in sediments from urban areas they generally depend on the intensity of traffic present [7]. During rainfall, these sediments are transported by river runoff to receiving waters, where it has a great impact on water quality [8,9].

Some metals, in small amounts, are essential for the functioning of organisms and plants (Fe, Mn, Zn, Cu and Mo)I; in high amounts, they can be dangerous and harmful.

There are also metals that have no function and are toxic in small quantities, such as Hg, Pb or Cd [10]. Relevant studies have shown that potentially toxic elements (PTEs) in road dust have negative effects on human health, such as poisoning, cancer, respiratory diseases and even psychological diseases [11–13]. Additionally, heavy metals are persistent in the environment and bioaccumulate.

Generally, the predominant metals on roads are Zn, which comes from vehicle tires [14,15], and high concentrations of Pt, Cd, Cu, Pb and Zn have been found on roads with heavy traffic [16].

In recent studies, the importance of urban dust resuspension as an important source of diffusion has been investigated. Resuspension was found to be a culprit in PM10, PM2.5 and PM1 emissions [17,18]. These particles, and even particles smaller than 100 μm in diameter, are easily resuspended into the air by passing wind or motor vehicles [19,20]. These small particles are easily inhaled by humans and pose a risk factor for humans [21].

Various indices are used to assess the environmental risk caused by exposure to heavy metals in street dust [19,22–24]. They are very suitable tools that allow for knowing the situation for acting in case of risk. Color [25] and magnetic [26] measurements can be used as a low-cost technique "proxy" indicator of contamination of potentially toxic elements.

In recent years, the study of the concentration of heavy metals in street dust in cities has aroused growing interest and, thus, has been studied in highly populated capitals such as Mexico City [6], Hong Kong [27], Dakha [28], Istanbul [29] and Islamabad [30]. In Spain, Madrid was studied [24], and the city of Murcia [31] which is very close to Cartagena, was also studied.

In the city of Cartagena, there are no studies on urban dust pollution; however, due to its location near the mining industry La Union, its historical and important industrial activity, population growth and the increase in traffic in recent years, it seems interesting to carry out a study.

The initial hypothesis is that the urban dust in the city of Cartagena contains heavy metals, among other pollutants. Also, it is believed that the concentrations of these elements rely upon diverse factors such as traffic intensity and that the primary source of emission of heavy metals is motorized vehicles, without dismissing other sources. On that basis, the main objective of this research was to identify pollution hotspots caused by heavy metals in the urban dust of Cartagena city. For this purpose, the following partial objectives are proposed:

- Know the concentration of heavy metals.
- Compare the content of heavy metals in Cartagena with respect to other cities.
- Relate the concentration of heavy metals to the color and magnetism of urban dust in order to use them as a "proxy" method to determine the level of contamination.
- Relate dust pollution to traffic density.
- Apply geoenvironmental indexes.
- Elaborate metal concentration distribution maps and their geoambiental indexes.
- Establish an improvement proposal to improve the environmental quality of the city.

## 2. Materials and Methods

### 2.1. Study Area

The research site was located in Cartagena (Murcia). Cartagena has a population of 214,000 inhabitants and is located on the coast of the Mediterranean Sea. Samples were also taken in the neighborhoods of Los Mateos and Santa Lucia, on the outskirts of Cartagena, because they were close to the emissions of Zn or Pb factories, as well as a fertilizer factory, which is responsible for Cu, Pb, Ni and Cd emissions.

The city of Cartagena has a warm, arid Mediterranean climate. Its position close to the sea softens the temperatures, achieving an average annual temperature of 20 °C. In the coldest month, it has an average of 12 °C, while in the hottest month, it reaches an average of 28 °C. As for rainfall, it hardly exceeds 300 mm per year. In addition, the predominant

winds in the winter months come from the southwest while, in the other stages of the year, winds from the east and southeast predominate.

### 2.2. Sampling System

The design that was carried out was systematic. Samples were collected every 150 m, obtaining a total of 88 urban dust samples (Figure 1). Sampling was carried out in March 2023 during a period of low rainfall. Most of the samples were taken on roads or parking lots, places close to vehicle traffic, although samples were also collected on sidewalks and in parks. Samples were collected over a 1 m² area. Dust was swept over the entire surface using brushes. The collected dust was sieved with a 1 mm sieve (non-metallic) and finally stored in small plastic containers.

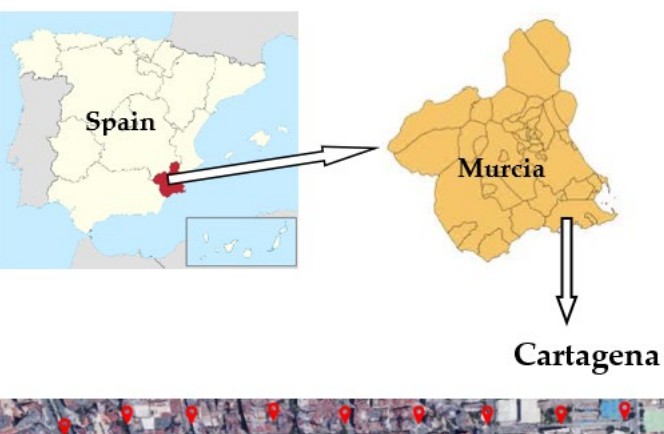

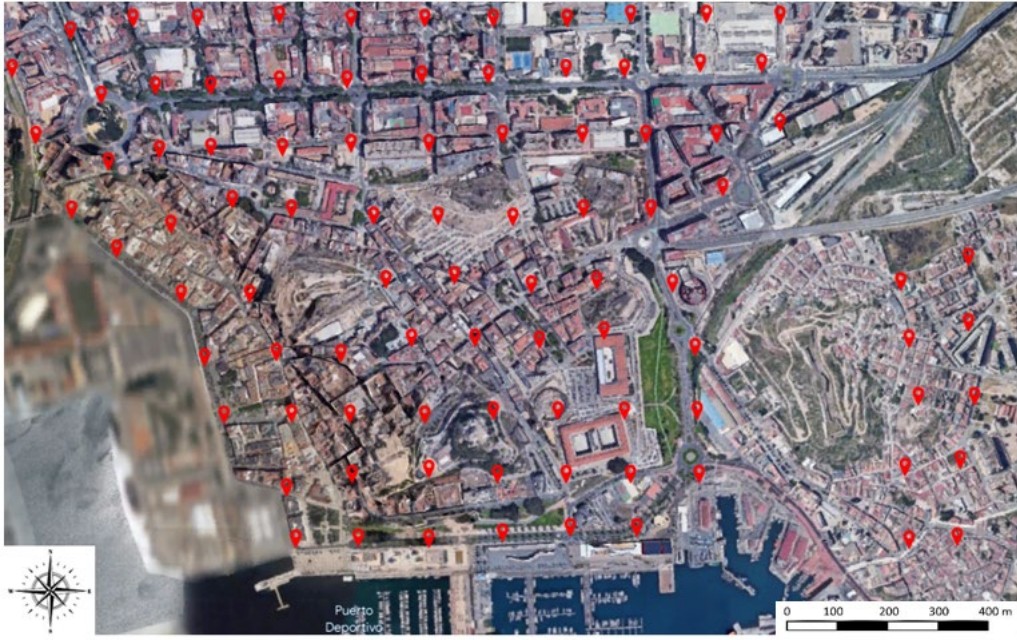

**Figure 1.** Study area and location of sampling sites. Cartagena city.

### 2.3. Sample Analysis

The color of the samples was estimated with the Munsell color keys [32] and then grouped into two different categories: Light (value between 5–6) and dark (value between 1–4). Most of the samples have the same 10YR HUE. To know the degree of magnetism, the same amount of powder was taken from all the samples, and using a magnet, the traces of metal dragged by the magnet were semi-quantified. In this way, three categories of magnetism level were made, classified by (+): for (+), low; for (++), medium; and for (+++), high.

In the absence of data on traffic intensity in the urban area of the city of Cartagena, a study relating noise to traffic [33] was used, where the streets with more noise were

considered to be of high traffic intensity, those with less noise were of medium intensity and, finally, those that did not appear were considered to be of low traffic intensity.

The samples were ground with an agate mortar and passed through a 50 μm sieve prior to acid digestion with aqua regia ($HNO_3$/HCl, 1:3) in a microwave oven at 220 °C for 1 h. Inductively coupled plasma mass spectrometry (ICP-MS) model Agilent 7900 was then performed to determine the total concentration of the following elements: Cd, Cu, Cr, Ni, Pb and Zn. All the chemicals used were of the highest purity available. High-quality water, obtained using a Milli-Q system (Milli-pore, Bedford, MA, USA), was used exclusively. Standard solutions (1000 μg mL$^{-1}$) of copper, lead, nickel, zinc, cadmium and chromium were purchased from Panreac (Barcelona, Spain) and diluted as necessary to obtain working standards. All elements were determined simultaneously with a multi-elemental pattern [34].

A certified standard reference material was analyzed for its element content (SRMSan Joaquin Soil), and duplicate samples were analyzed simultaneously to provide quality control. The standard deviation (2.5–3%) was calculated and can be considered satisfactory for environmental analysis. Recoveries of 93–102% for Zn, 95–101% for Pb, 91–98% for Cu, 94–99% for Cr, 98–102% for Ni and 95–101% for Cd were obtained.

### 2.4. Environmental Pollution Index

#### 2.4.1. Contamination Factor

The contamination factor (CF) describes the level of contamination in street dust for a given metal. It is calculated by the ratio between the concentration of a heavy metal (Cn) and its background value (Cbn) [35].

$$CF = Cn/Cbn$$

Based on the obtained CF results, the level of heavy metal contamination has been set according to CF < 1, low; $1 \leq CF \leq 3$, moderate; $3 \leq CF \leq 6$, considerable; and $CF \geq 6$, very high.

#### 2.4.2. Pollution Load Index

The pollutant load index (PLI) evaluates the heavy metal contamination status of sediment samples. It is a direct geometric mean of the CF and is expressed as [36]:

$$PLI = (CF_{Me1} \times CF_{Me2} \times CF_{Me3} \times \cdots \times CF_{Men})^{1/n}$$

PLI values < 1 indicate the absence of heavy metals, while PLI > 1 shows the presence of heavy metal contamination.

#### 2.4.3. Enrichment Factor

The enrichment factor (EF) originally evaluated the level of metal contamination in air [37], but now it is also used for dust and soil. In addition, EF serves to know the degree of human influence, as it helps to distinguish heavy metals of anthropogenic origin from those coming from natural sources [38]. EF was calculated as follows:

$$EF = \frac{\left(\frac{C_n}{C_{Ref}}\right)_{Sample}}{\left(\frac{C_n}{C_{Ref}}\right)_{Background}}$$

Cn is the concentration of an element n, and Cref is the reference concentration of element n. For the values of background concentrations, the values of asphalt will be used.

Based on EF, the metal enrichment level categories are as follows [39]: $EF \leq 2$, low to minimal enrichment; EF = 2–5, moderate enrichment; EF = 5–20, considerable enrichment; EF = 20–40, high enrichment; and EF > 40, very high enrichment. EF values close to 1 mean

that they are of natural origin; on the other hand, when EF > 10, they are of anthropogenic origin [40].

### 2.4.4. Geo-Accumulation Index

It is a quantitative measure of the extent of metal contamination in the soil studied using the geo-accumulation index (Igeo) proposed by Muller (1969) [41]. The index (Igeo) of heavy metals considers small variations in the background value using a factor of 1.5 (K-factor) and is expressed as:

$$Igeo = log2\frac{C_n}{K \times C_{bn}}$$

where Cn is the metal concentration (n), Cbn is the metal background concentration (n) and K is the correction factor that compensates for the background data due to lithospheric effects, which is generally defined as 1.5.

Igeo values can be interpreted as follows [41]: Igeo < 0 (not polluted), 0–1 (not polluted to moderately polluted), 1–2 (moderately polluted), 2–3 (moderately to highly polluted), 3–4 (highly polluted), 4–5 (highly polluted to very highly polluted), and Igeo > 5 (very highly polluted).

### 2.4.5. Potential Ecological Risk Index

Risk Potential Index (RI) integrates the potential ecological risk factors of each metal and associates its ecological and environmental effects with its toxicology [42]. RI is expressed as:

$$Er^i = Tr^i \times CF^i$$
$$RI = \sum Er^i$$

where, $CF^i$ is the contamination factor of each metal, $Er^i$ is the ecological risk potential of each metal, $Tr^i$ is the toxic response coefficient [43] and, finally, RI is the ecological risk index [43].

The RI can be interpreted based on the following categories: RI < 150, low ecological risk; $150 \leq RI \leq 300$, moderate ecological risk; $300 \leq RI \leq 600$, considerable ecological risk; and RI > 600, very high ecological risk.

### 2.5. Statistical Analysis and Mapping

The different statistical studies, both for basic statistical calculations and for index calculations, were carried out using Microsoft Excel v2305 (Microsoft Corporation, Redmon, WA, USA). The Surfer v23 program (Golden Software LLC, 1983, Golden, CO, USA) was used to make the distribution maps.

## 3. Results

The values of means, medians, maximums, minimums and standard deviation of the concentration of six metals of urban dust (n = 88) are represented in Table 1. As no previous studies have been conducted on urban dust in the city of Cartagena, no background values are available for comparison with those obtained in this study; therefore, they will be compared with studies in other areas, such as those conducted in the soils of the city of Murcia.

**Table 1.** Statistical values of the concentrations (mg kg$^{-1}$) of the elements analyzed in the city of Cartagena.

|  | Cd | Cr | Cu | Ni | Pb | Zn |
|---|---|---|---|---|---|---|
| Mean | 4.1 | 82.7 | 248.9 | 47.7 | 227.2 | 672.5 |
| Median | 3.4 | 67.6 | 201.5 | 30.9 | 184.3 | 624.1 |
| Max. | 21.7 | 848.9 | 1903.8 | 701.6 | 852.4 | 2253.4 |
| Min. | 0 | 0 | 15.3 | 12.5 | 11.6 | 63.6 |
| SD | 3.72 | 94.71 | 251.33 | 96.21 | 158.54 | 328.12 |

In the streets of Cartagena, it has been found that the concentration of the 11 metals has a considerable variation so the mean values range from 4.08–25,365.51 mg kg$^{-1}$ (Table 1), corresponding to Cd and Fe. Also, the mean levels of heavy metals in dust can be arranged in the following order: Zn > Cu > Pb > Cr > Ni > Cd.

On the other hand, there are metals whose average concentration, according to the thresholds proposed [44], exceeds the maximum permitted values, such as As (29.88 mg kg$^{-1}$), Cd (4.08 mg kg$^{-1}$), Pb (227.05 mg kg$^{-1}$), or even reaching values where an investigation is mandatory, as is the case of Zn (672.05 mg kg$^{-1}$). Several studies [31,36] have shown which heavy metals have an anthropogenic origin, namely Cd, Cr, Cu, Ni, Pb and Zn. These heavy metals will be studied in more depth in this paper.

The relationship between the different variables of magnetism, traffic intensity and color of the samples is shown in Table 2. It is observed that the relationship between magnetism and color is as expected since the samples with low and medium magnetism have a majority of light-colored samples, with 81.8% and 83.3% of light-colored samples, respectively, on their magnetism level. Meanwhile, samples with high magnetism levels provide more dark-colored samples, with 55.5%. As for the relationship between traffic and color, there was no clear relationship since, at all levels of traffic intensities, the light color of the samples predominate.

**Table 2.** Relationship between color of samples according to magnetism and traffic density.

|  |  | n | Light | Dark |
|---|---|---|---|---|
| Magnetism | Low | 22 | 19 | 3 |
|  | Medium | 30 | 25 | 5 |
|  | High | 36 | 16 | 20 |
| Traffic Intensity | Low | 54 | 36 | 18 |
|  | Medium | 12 | 9 | 3 |
|  | High | 22 | 15 | 7 |

When comparing traffic intensity with magnetism (Table 3), there is a correlation between them since the streets with less traffic density have practically the same samples of high, medium and low levels of magnetism. Meanwhile, in the streets with medium and high traffic density, a high level of magnetism predominates.

**Table 3.** Relationship of samples between magnetism and traffic intensity.

|  |  | Magnetism | | |
|---|---|---|---|---|
|  |  | Low | Medium | High |
| Traffic intensity | Low | 16 | 19 | 19 |
|  | Medium | 2 | 3 | 7 |
|  | High | 5 | 8 | 9 |

### 3.1. Zinc

Most of the presence of Zn in the environment is due to human activities. Zinc, lead and cadmium ore refining, mining, steel production, as well as coal combustion and other wastes, can emit Zn into the environment [14,15].

It is commonly used as a coating for steel and iron to prevent rust or corrosion. It is considered a metal of low toxicity since it is necessary in the human diet in small amounts. However, its exposure in large quantities and for a certain period of time can cause stomach cramps, nausea and vomiting, which, in turn, can produce anemia, damage to the pancreas and a decrease in the beneficial type of cholesterol [45].

Zn was one of the metals with the highest concentration in street dust in Cartagena. The average Zn concentration is 672.5 mg kg$^{-1}$ (Table 1), reaching a maximum of 2253.4 mg kg$^{-1}$.

It is observed that Zn concentrations in the streets of Cartagena are in the range of 1200–300 mg kg$^{-1}$ (Figure 2A) in most of the streets, except in two points where concentrations exceed 2000 mg kg$^{-1}$. These areas are residential and close to a school, which is a problem for the health of people living nearby.

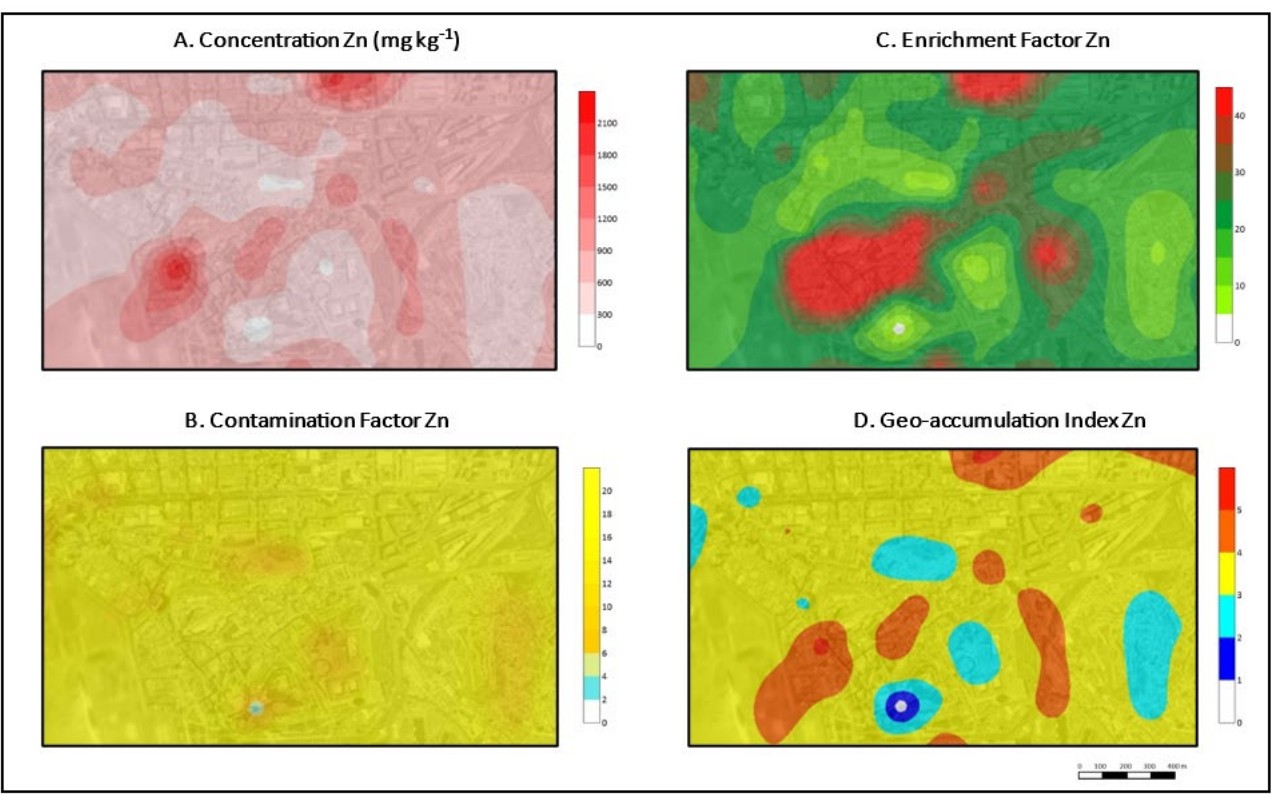

**Figure 2.** Spatial distribution of concentration and geoambiental index of Zn in Cartagena.

The mean EF of Zn with respect to the background (asphalt) was 23 (Figure 2B), which represents a high enrichment in the categories defined above. Some areas exceed 39, even reaching values of 92, but in most of the sampled areas, EF values between 10–30 are found. Except for some specific zones, most of the map (Figure 2B) shows that the EF values exceed 10, which means that the Zn contamination comes from anthropogenic sources [40].

The values shown (Figure 2C) are high, practically in the entire sampling area; according to the defined categories, this represents a very high CF, reaching a maximum of 65, although the average value is 19.

The Igeo remained with values between 3–4 in most of the area, as shown in yellow in Figure 2D. Having an average Igeo of 3.5 implies that the area is highly contaminated with Zn, based on the previous categories [41]. Some zones exceed the value of 5, which qualifies them as highly contaminated zones.

When comparing Zn concentrations with other variables such as magnetism, traffic and color (Figure 3), a relationship was observed between Zn concentrations and magnetism. In both light and dark colors, the concentration decreases as the level of magnetism decreases. Also, it is observed that Zn concentrations are higher in the dark color samples. As for the traffic, in the light-colored samples, a decrease in the Zn concentration is observed as the traffic intensity decreases, while this same phenomenon does not occur in the Ddark-colored samples.

The main source of Zn emissions comes from the wear of tires, brake pads, oils, greases and lubricants of motor vehicles [46], so the highest concentrations of Zn will be found in the streets and in those with the highest traffic intensity, although this is only found in the light-colored samples. In the dark-colored samples, there is more Zn content in the streets with less traffic density.

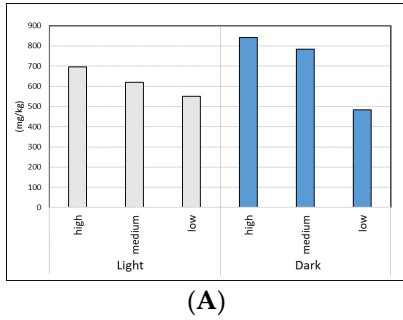

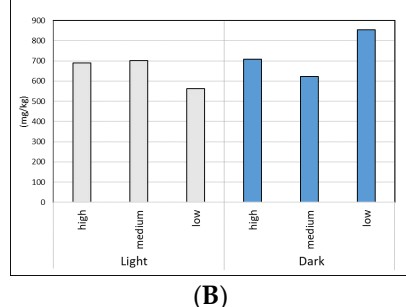

**Figure 3.** Mean Zn concentration according to color and magnetism (**A**) and color and traffic intensity (**B**).

The solubility of Zn is very low in soils, according to Tiller [47], so one of the only ways to remove Zn from streets is by rain and stormwater, which can wash Zn into the drainage system. This would cause the Zn to move from being in the streets to being in the water, which is a problem for the water quality of the receiving waters [8,9]. Since it does not rain in Cartagena, streets and sidewalks are not cleaned by rain. Frequent washing of streets is highly recommended to avoid the accumulation of dirt in the streets [48].

### 3.2. Lead

The average Pb concentration in Cartagena was 227.2 mg kg$^{-1}$ (Table 1). Decades ago, vehicle emissions were the main source of Pb emissions, but with the elimination of Pb in fuels in 1970, emissions decreased drastically [46].

In a large part of the area, Pb concentrations do not exceed 300 mg kg$^{-1}$ (Figure 4A). Regarding the thresholds proposed [44], an investigation is recommended.

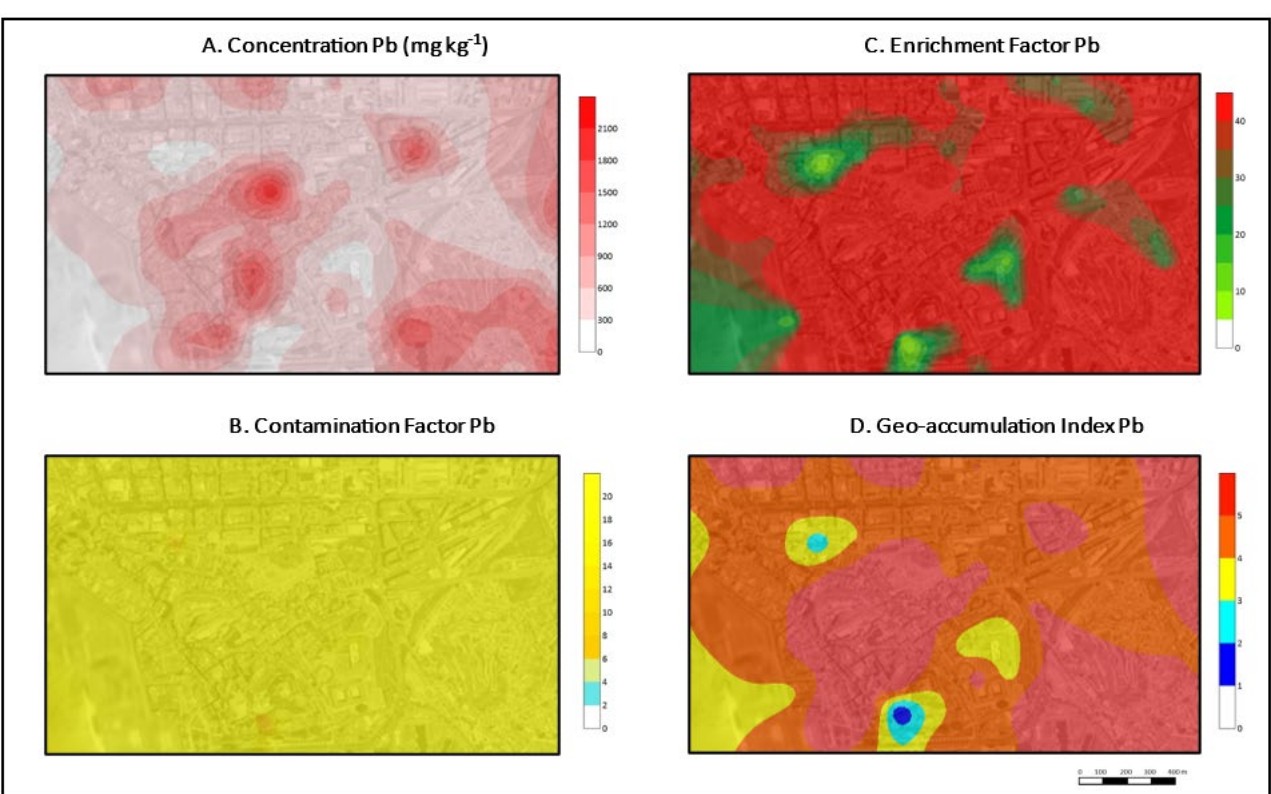

**Figure 4.** Spatial distribution of concentration and geoambiental index of Pb in Cartagena.

The EF of Pb in the dust of the city of Cartagena, supported by the background (asphalt), is 53, being an area of very high enrichment. High EF of Pb in most of the samples

(Figure 4B) means that the contamination comes from human activity and represents a significant degree of contamination.

Like the EF, the Pb contamination factor in the city of Cartagena shows very high values in practically the entire sampling area, being greater than 20 (Figure 4C). The mean CF is 64, which means a very high Pb contamination, exceeding the last category described above.

In this case, the Pb Igeo is higher than that shown for Zn (Figure 4D), with an average value of 4.8; according to the Igeo categories, this represents an area highly contaminated by Pb.

When comparing the different variables, it is observed that there is the same relationship as there was with Zn regarding magnetism, color and Pb concentration (Figure 5). In the light- and dark-colored samples, the Pb concentration decreases along with the magnetism. On the other hand, when relating traffic to concentration, it is observed that in the light-colored samples, the Pb concentration increases as the degree of magnetism decreases, the opposite phenomenon to Zn. As for the dark-colored samples, it is worth mentioning the streets with medium traffic intensity, where the Pb concentration is maximum.

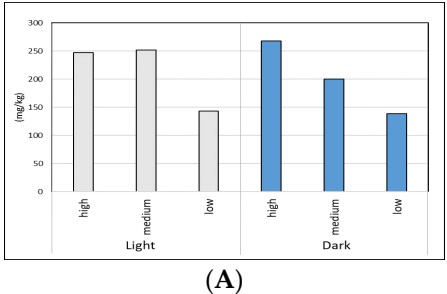 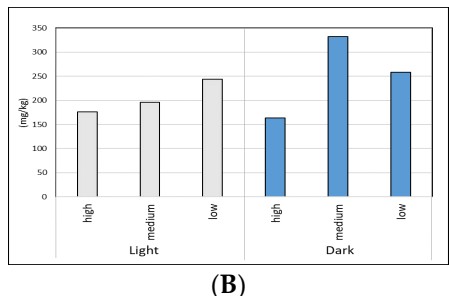

(**A**)          (**B**)

**Figure 5.** Mean Pb concentration according to color and magnetism (**A**) and color and traffic intensity (**B**).

*3.3. Copper*

The anthropogenic sources of Cu include mining and its derivatives, the electricity industry, agriculture, the steel industry and sewage sludge. Most of the Cu composites found in the air, water, sediment, soil and rocks are firmly attached to dust or to other particles in suspension [49]. Cu is qualified as a non-carcinogenic metal in humans. In addition, it is necessary to ingest small amounts of Cu, although ingesting too much Cu can cause vomiting, abdominal pain or nausea [50].

The average concentration of Cu in street dust in Cartagena is 248.9 mg kg$^{-1}$ (Table 1).

Observing the Cu map (Figure 6A), it can be seen that it is quite uniform, except for one point where the Pb concentration reaches a maximum of 1903.8 mg kg$^{-1}$. According to the thresholds [44], this is an area where it is advisable to investigate its Cu content. Quite a lot of copper near the port, with possible origin related to port activity.

EF of Cu in street dust in Cartagena (Figure 6B), supported by the background (asphalt), is 16, which represents a moderate enrichment. In this case, there are some areas where the enrichment factor is less than 10, which implies that the Cu does not come from anthropogenic origin.

In this case, the CF in the streets of Cartagena for Cu is 13.8, implying a very high Cu contamination (Figure 6C). On this occasion, CF values of less than 6 can be seen in certain areas, which was not the case for the CF of Zn or Pb.

The Cu Igeo map (Figure 6D) shows areas that are not painted, which means that their Igeo value is less than 0 and, therefore, they are areas that are not contaminated by Cu. The mean Cu Igeo value is 2.72; according to the categories described above, this means a zone moderately to highly contaminated by Cu.

As with Zn and Pb, the same relationship of magnetism with Cu concentrations is also true for Cu in both colors (Figure 7). As for traffic intensity, in the light-colored samples, the

Cu concentration is high in streets with high traffic intensity, whereas, in the dark colors, the highest concentrations are found in streets with high and low traffic density.

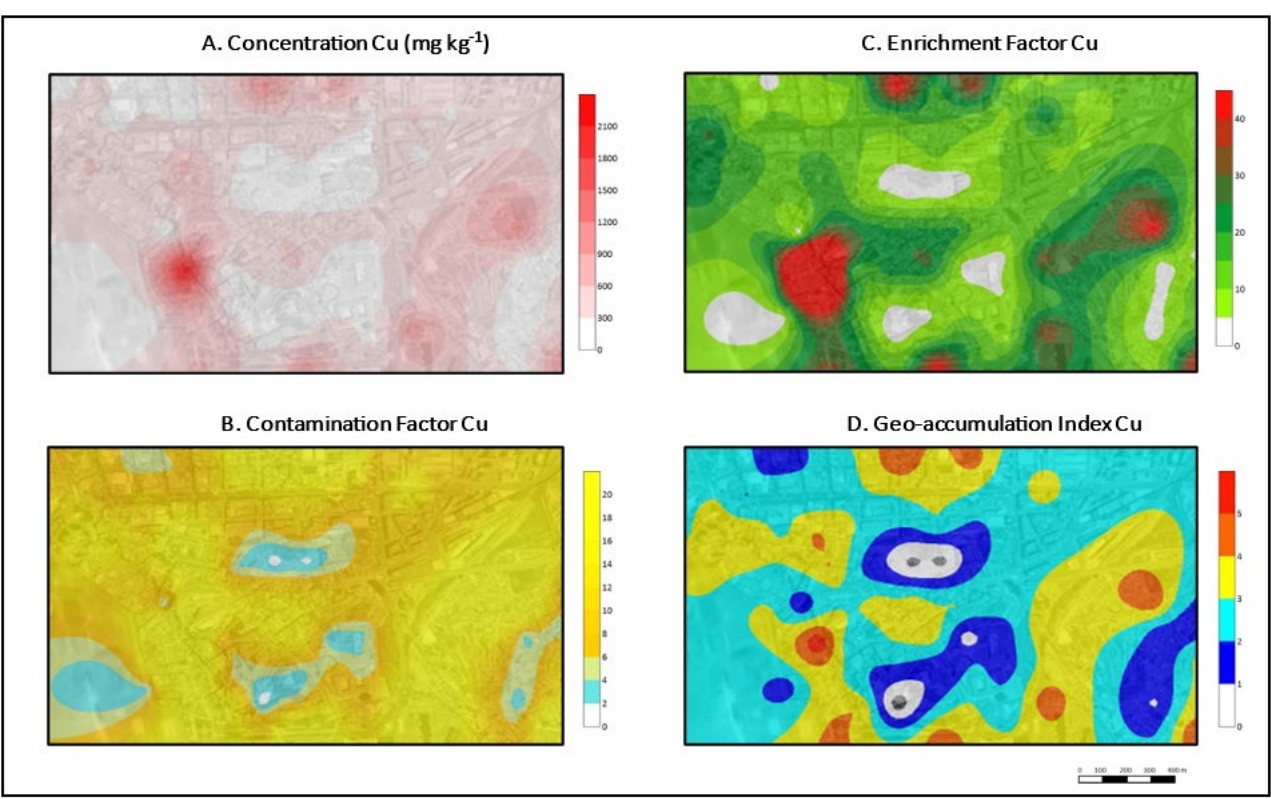

**Figure 6.** Spatial distribution of concentration and geoambiental index of Cu in Cartagena.

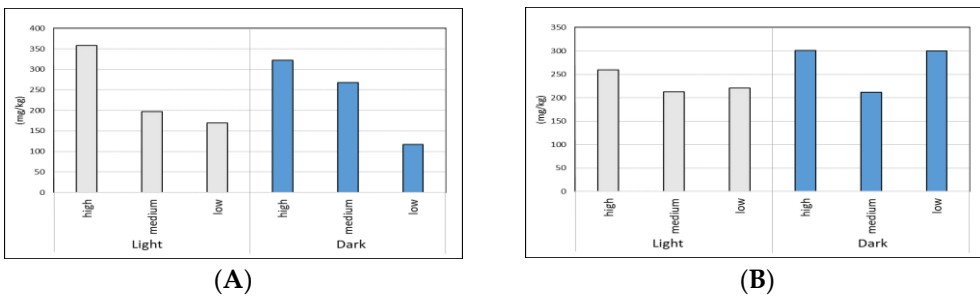

**Figure 7.** Mean Cu concentration according to color and magnetism (**A**) and color and traffic intensity (**B**).

### 3.4. Nickel

Nickel is released into the environment by industries that manufacture alloys and other compounds, fertilizers, power plants and garbage incinerators [6]. It usually takes days for nickel to be removed by air, and if it adheres to small dust particles, it can take months to be deposited in the soil. In addition, it is considered an essential element for soil organisms. Exposure to Ni causes dermatitis, asthma attacks, gastric irritation and ontological processes [6].

The mean Ni in the samples was 47.7 mg kg$^{-1}$ (Table 1). According to the thresholds [44], Ni is below the maximum allowed.

Observing the map of Ni concentrations (Figure 8A), uniformity can be seen in a large part of the map, where the concentrations do not exceed 100 mg kg$^{-1}$, with the exception of two specific areas, where they reach 700 mg kg$^{-1}$ of Ni, which will also be hotspots for other heavy metals such as zinc.

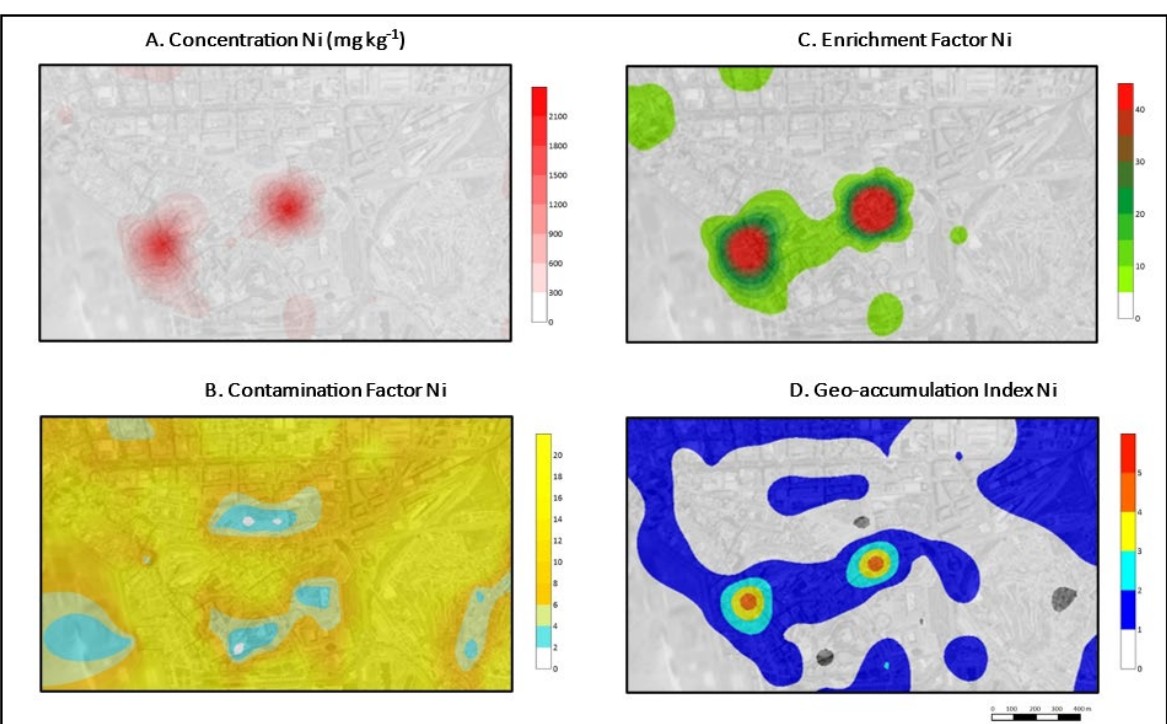

**Figure 8.** Spatial distribution of concentration and geoambiental index of Ni in Cartagena.

The EF value in the streets of Cartagena is 6, which represents a zone of considerable Ni enrichment. As in the Ni concentration map, in the Ni EF map (Figure 8B), the values are the same in most of the map (EF < 5), except for two points where they exceed the value of 40 EF, indicating that the Ni emission in these areas comes from an anthropogenic origin. Therefore, in the other points, the Ni content is of natural origin.

The CF in the streets of Cartagena is 4.23, which implies a considerable level of contamination and is the lowest of the six metals. As in the Ni concentration map (Figure 8C) and in the EF map (Figure 8B), the two areas where the CF values are higher are the same as in the previous two maps.

When analyzing the map (Figure 8D), we observe non-contaminated zones, as in the Cu Igeo map. The Ni Igeo is 1, which represents a moderately contaminated zone, and the same pattern of the previous Ni maps is again fulfilled with respect to the two most contaminated zones.

Examining the relationship between Ni concentration (mg kg$^{-1}$) and magnetism, the concentration decreases along with the level of magnetism in both colors. In the dark samples, the concentrations are maximum at both the magnetism level (Figure 9A) and traffic density (Figure 9B). In the light-colored samples, at all traffic density levels, the Ni concentration is practically the same.

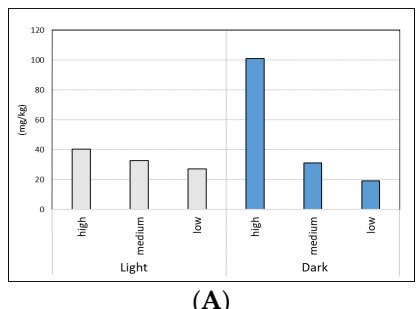

**(A)**

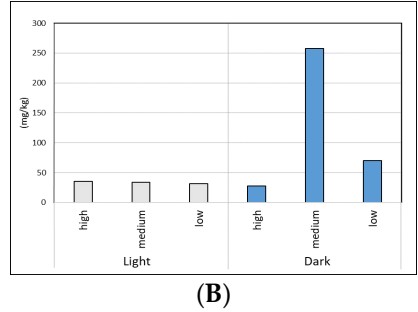

**(B)**

**Figure 9.** Mean Ni concentration according to color and magnetism (**A**) and color and traffic intensity (**B**).

### 3.5. Chromium

Cr can be found in soil, air and water after its release from vehicle braking, coal burning from power plants and oil, textile production and the manufacture of chromium-based products. Regarding the origin of the Cr found in street dust, the reference consulted relates it to the plating industry, steel alloys, other automobile parts and street infrastructure [51].

Cr enters the environment in the form of Cr III and Cr VI. Cr III is necessary for life, while Cr VI is carcinogenic, having effects on the respiratory, gastrointestinal, immune and reproductive systems [52].

The mean Cr concentration found in the streets of Cartagena is 83.7 mg kg$^{-1}$ (Table 1).

Analyzing the Cr map (Figure 10A), it can be seen that in most of the city, it does not exceed the limit proposed [44], although there is an area where the Cr concentration (848.9 mg kg$^{-1}$) exceeds the intervention level.

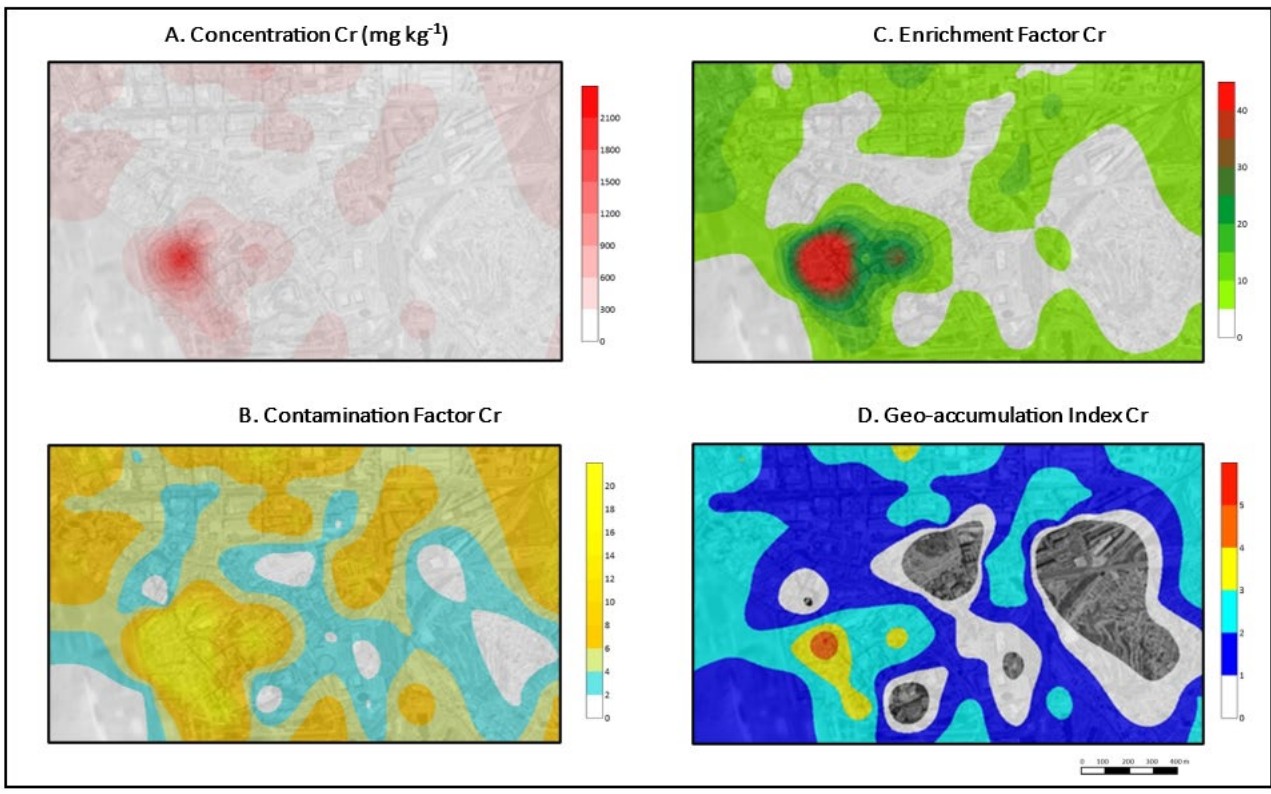

**Figure 10.** Spatial distribution of concentration and geoambiental index of Cr in Cartagena.

As for the EF, its average value is 7.8, which is considered a zone of considerable Cr enrichment. One zone exceeds 40 EF (90) (Figure 10B), making it a zone of very high Cr enrichment.

The contamination factor in the streets of Cartagena is 6.28; according to the previous categories, this represents a high Cr contamination zone. The zone with the highest CF is located in the southwest zone (Figure 10C), a commercial zone where many people pass through.

The Cr Igeo value is 1.4, indicating a moderately contaminated area. There are several areas where the Igeo is less than 0 (Figure 10D), indicating areas not contaminated by Cr; meanwhile, the western area has the highest Igeo.

Higher Cr concentrations were found in the dark samples when compared to magnetism (Figure 11). In both colors, the decrease in Cr concentration along with magnetism is still fulfilled, as in the metals studied previously. When comparing with traffic, the samples with the highest concentration are the dark-colored and low-traffic intensity samples.

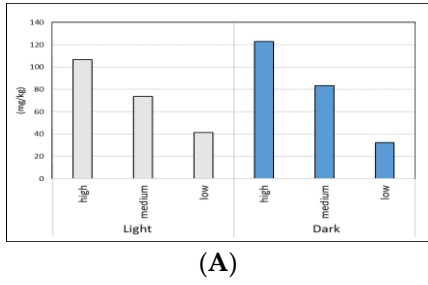
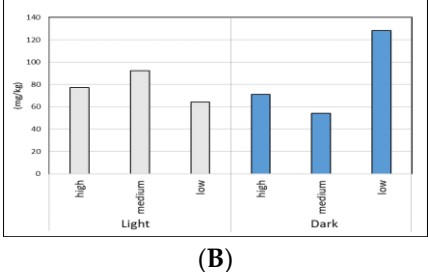

(**A**)                                                           (**B**)

**Figure 11.** Mean Cr concentration according to color and magnetism (**A**) and color and traffic intensity (**B**).

## 3.6. Cadmium

Cd enters the environment through mining and refining of non-ferrous metals, fertilizers, fossil fuel combustion and waste incineration. The mobility of Cd and its components depends on several factors, such as pH and organic matter in the environment, becoming immobile by binding strongly to organic matter. Many health agencies have determined that Cd is a carcinogenic element that causes lung cancer in humans [53].

The areas with the highest Cd concentration are located in the northeastern part of the city (Figure 12A), in areas with high traffic intensity, reaching values of 21.7 mg kg$^{-1}$, exceeding the intervention level proposed [44] for Cd.

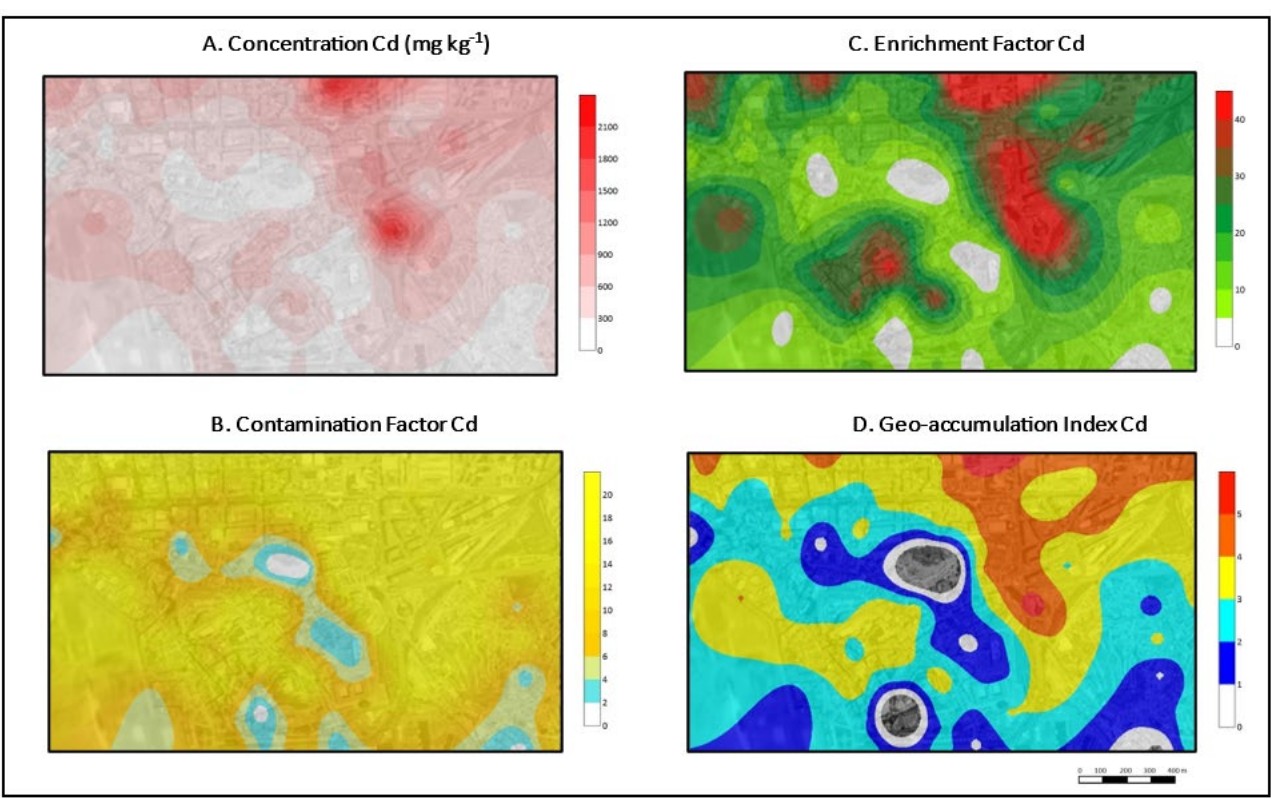

**Figure 12.** Spatial distribution of concentration and geoambiental index of Cd in Cartagena.

The EF of Cd in the streets of Cartagena, based on the background value (asphalt), is 18, being an area with a considerable enrichment of Cd. On this occasion, the maximum values are found in the central-northern part of the sampling area, exceeding 40 EF (82), being areas with a very high enrichment of Cd.

The CF of Cd is 15.1, assuming an area highly contaminated by Cd. Observing the map (Figure 12C), it can be seen that the highest CF values (80) are found throughout the

north, while in the central and southern areas, CF values are lower, reaching very close to 0 in some cases.

The average Igeo in Cartagena is 2.7, being a moderately to highly polluted area. As in the case of CF, the highest Igeo values are found in the northeast (Figure 12D), while in the south and center, the Igeo values are lower, and there are areas with no Cd contamination.

On this occasion, the relationship that was present in the 5 other metals is not fulfilled in the samples that are light colored (Figure 13). The light- and dark-colored samples do not follow the same pattern in relating magnetism to Cd concentration. In the light-colored samples, the highest Cd concentrations are found at low levels of magnetism, and in the dark-colored samples, the Cd concentration decreases as magnetism decreases. In terms of traffic and concentration, in the light-colored samples, the concentration is higher in medium-traffic intensity streets; on the other hand, in the dark-colored samples, the concentration is higher in low-traffic intensity streets.

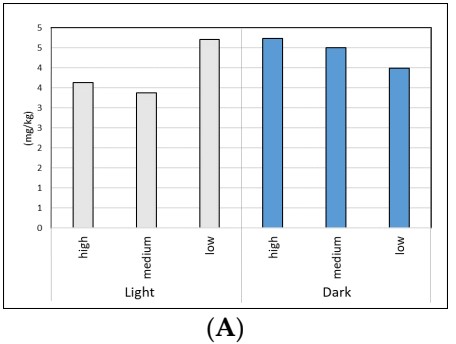 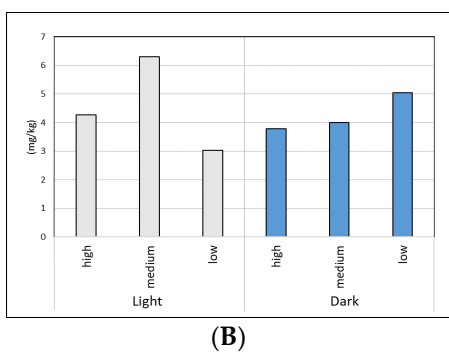

(**A**)  (**B**)

**Figure 13.** Mean Cd concentration according to color and magnetism (**A**) and color and traffic intensity (**B**).

*3.7. Ecological Risk Potential Index (RI) and Pollutant Load Index (PLI)*

Both indices depend on the pollution factor (CF); therefore, it is an important variable when observing the results. The PLI shows a direct contribution of CF from each of the metals. The PLI, in a large part of Cartagena, is greater than 1, meaning that in a large part of the city of Cartagena, there is contamination by heavy metals. The only areas where there is no heavy metal contamination are located in Monte Sacro and Plaza Puerta de la Villa.

As in other maps of various metals such as Zn (Figure 2), Ni (Figure 8) and Cr (Figure 10), where their highest concentration was found at the same point, to the southwest, in this case, the highest PLI value (57) is found in the same place, as this point is a tourist-commercial area where many people transit throughout the day.

As for the RI, it has an average value of 841; according to the categories described above, this represents a very high ecological risk. The map (Figure 14) shows that the highest RI values are found in the northeast of the city, corresponding to streets and areas at the entrance to the city; therefore, these are areas with high traffic density. The northern part of the city is the least polluted, and it is a newer area with larger avenues and smoother traffic flow.

The hotspot is in the southwest of the city, the most polluted area in global terms, as indicated by the RI and PLI indices. This is an area in the old part of Cartagena, with dense traffic, where vehicles drive very slowly due to the presence of traffic lights that force vehicles to stop and start constantly. This area is also next to the military arsenal and the port and, therefore, is influenced by the presence of ships. The hotspot is in the vicinity of the port, where port activities generate a variety of pollutants [54].

The neighborhoods of Los Mateos and Santa Lucía, located in the southeast of the city and sampled due to their proximity to industrial and mining areas, do not seem to be a hotspot of contamination, as their values are in the general trend of the city. In the past, they did present serious air pollution problems.

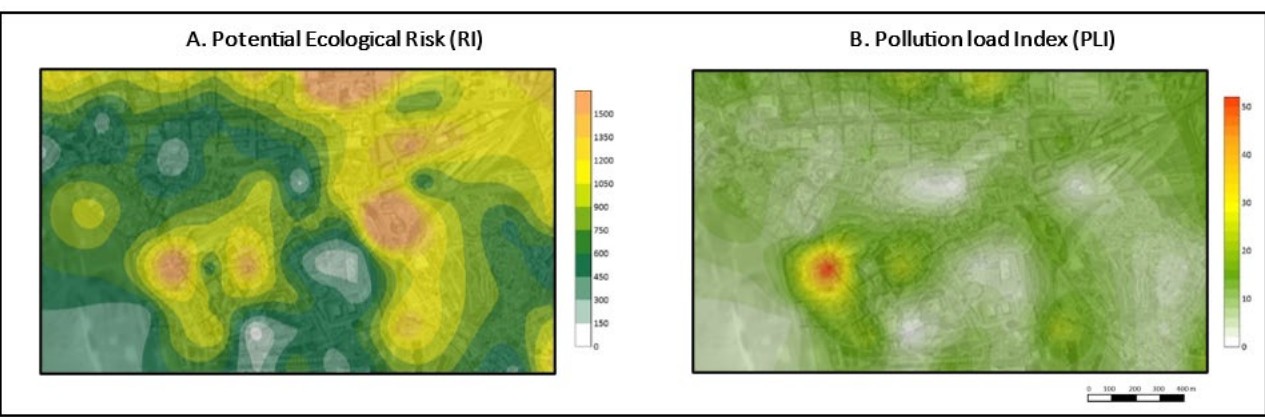

**Figure 14.** Spatial distribution Pollution Load Index and Potential Ecological Risk in Cartagena.

## 4. Discussions

The values obtained in urban dust have been compared with cities in other parts of the world, where studies similar to the present one have been carried out.

It can be seen that, in Cartagena, the concentrations of Cr, Ni and Zn are similar to those found in the city of Murcia [31], while the content of Cd, Cu and Pb is higher. When compared with Madrid [24], the Pb content stands out since it exceeds the content present in Cartagena; as for the other metals, there are no major differences. In general, the concentrations of metals in Cartagena are higher than in the cities shown in Table 4, except in Karachi [55], Pakistan, where large differences are observed between the two.

**Table 4.** Concentrations (mg kg$^{-1}$) of the elements analyzed in different cities.

| City | Inhabitants | Cd | Cr | Cu | Pb | Ni | Zn | Reference |
|---|---|---|---|---|---|---|---|---|
| Cartagena | 214,000 | 4.1 | 82.7 | 248.9 | 227.0 | 47.7 | 672.0 | Present work |
| Murcia | 439,712 | 0.5 | 118 | 201 | 177.0 | 51.0 | 653.0 | [31] |
| México city | 8,855,000 | - | 126.0 | 97.0 | 206.0 | 51.0 | 321.0 | [6] |
| Madrid | 6,000,000 | 1.25 | 100 | 411 | 290 | 42 | 898 | [24] |
| Ahvaz | 1,185,000 | 5.6 | 107.4 | 187.1 | 189.1 | - | 93.6 | [23] |
| Lublin | 342,039 | 5.1 | 86.4 | 81.6 | 44.1 | 16.5 | 241.1 | [56] |
| Karachi | 14,910,000 | 62.3 | 148.1 | 332.9 | 426.6 | 389.7 | 4254.4 | [55] |
| Backgroud | | - | 13.0 | 18.0 | 4.3 | 11.0 | 35.0 | Present work |

Zn was one of the metals with the highest concentration in street dust in Cartagena as well as in other cities such as Mexico [6], Lublin [56], Madrid [24] and Murcia [31].

The average Pb concentration in Cartagena was 227.2 mg kg$^{-1}$ (Table 1), slightly higher than in Murcia [31], Mexico City [6] or Singapore [57]. Decades ago, vehicle emissions were the main source of Pb emissions, but with the elimination of Pb in fuels in 1970, emissions decreased drastically [46]. In a large part of the area, Pb concentrations do not exceed 300 mg kg$^{-1}$ (Figure 3), being within the limits of countries such as Mexico or the United States (400 mg kg$^{-1}$) [6], but higher than the limit imposed on urban streets in Canada (140 mg kg$^{-1}$).

In this case, the Pb Igeo is higher than that shown for Zn (Figure 2D), with an average value of 4.8; according to the Igeo categories, this represents an area highly contaminated by Pb. When compared to cities such as Madrid [24], the Igeo Pb is lower but higher than cities such as Lublin [56].

Most of the Pb content in dust is emitted by vehicles, either from fuel, oil burning, brake wear, etc. [14,55,58]. Thus, it would be logical to think that the Pb concentration should be higher in high-traffic streets; however, in this case, the opposite occurs, both in

light- and dark-colored samples, the streets with lower traffic density are the ones with the highest Pb content. This suggests that in streets with low traffic intensity, vehicles drive slower, stop and park; therefore, this may be one of the reasons for their high Pb content or that the frequency of cleaning of streets with lower traffic density is lower than those with higher traffic density.

One source of Pb is the resuspension of Pb [58], where it can accumulate in places such as windows, sidewalks, balconies, etc. This aspect should be taken into account when studying the influence that Pb can have on health since it is a highly toxic element and has no function in any organism [10]. To reduce this risk, a cleaning system should be studied since the streets where traffic intensity is lower are mostly residential streets.

The average concentration of Cu in street dust in Cartagena is 248.9 mg kg$^{-1}$ (Table 1). Compared to other cities, its mean is higher than other cities such as Murcia [31], Beijing [33] or Madrid [24]. Moreover, it exceeds the limits in urban streets in Canada (63 mg kg$^{-1}$).

Anthropogenic sources of Cu include mining and its derivatives, the electrical industry, agriculture, the iron and steel industry and sewage sludge. However, in this case, the main source of Cu contamination comes from motor vehicles, specifically, the wear of brakes and brake pads [55].

The mean Ni in the samples was 47.7 mg kg$^{-1}$ (Table 1), the same as other contaminated cities [31,34,41].

The mean Cr concentration found in the streets of Cartagena is 83.7 mg kg$^{-1}$ (Table 1), which is lower than most of the cities shown in Table 2. When compared with the Cr limits in the streets of Mexico (280 mg kg$^{-1}$) and the United States (210 mg kg$^{-1}$) [6], it does not exceed them; on the other hand, it does exceed the limits established in Canada for Cr (64 mg kg$^{-1}$) [39].

The average Cd concentration in the streets of Cartagena is 4.1 mg kg$^{-1}$, which is higher than in the streets of cities such as Murcia, Beijing and Hong Kong.

In our study, the most polluted area was the area near the port. Port activity, including loading/unloading services, supplies, fuel, personnel transfer and repairs, can contribute to the presence of polluted dust in urban areas near ports. Dust in port areas may contain a variety of contaminants, such as Arsenic, Cadmium, Copper, Molybdenum, Lead, Zinc, organic matter, trace metals and polycyclic aromatic hydrocarbons (PAHs) [54].

Considering the color of the dust samples collected, the darkest ones correspond to a high magnetism but low traffic intensities, very slow traffic streets and the old city center, as was the case in the streets of Murcia studied by Marín et al. [31]. The analysis of the color of the urban dust revealed that the dark samples were the most contaminated; thus, the color can be considered a proxy for rapid diagnoses [25].

## 5. Conclusions

According to the results shown in the concentrations of urban dust from the streets of Cartagena, Zn was the metal with the highest concentration, requiring a mandatory investigation in the area. Following Zn, the other metals where an investigation is recommended are Cu and Pb. Meanwhile, Ni and Cr are within the permitted limits. When comparing the results obtained with several cities, the concentration averages of Ni and Cr are equal or lower than the values of the cities, while Zn, Cd, Cu and Pb are equal or higher in most of the cities.

When analyzing the different variables in the six metals studied, some relationships between them have been observed that can help to identify, in an approximate way, the places that may be contaminated by heavy metals. When classifying the samples by color and degree of magnetism, it was identified that the highest concentrations were found in the dark-colored samples, with the exception of Pb. Also, the relationship between magnetism and concentration of each metal is similar in all metals (regardless of the color of the sample) since the decrease of any of these variables affects the other in the same way. Likewise, magnetism can be an important and quite reliable "proxy" variable in identifying contaminated areas and hotspots.

Urban dust pollution is not reflected in traffic intensity, where the maximum concentration values were found, in most cases, on streets with little or medium traffic density. Therefore, this means that the main emissions may come from the braking and starting of vehicles since streets with less traffic tend to be residential and, therefore, are streets where people tend to brake often. Also, another factor to take into account is the classification of the streets. As we do not have data on traffic density, it has had to be approximated by a noise study; therefore, it would be better to reevaluate it when real traffic data is available.

The most important hotspot is located in the southwest of the city. It is the most polluted area in global terms, especially by Zn, Ni, Cu and Cr. It coincides with an area of the old part of the city where traffic circulation is very slow, with many traffic lights that force vehicles to stop and start constantly. This area, Calle Real, is adjacent to a military and marine area with the influence of the port and pollution from ships. The most polluted area was the area near the port. Port activity can contribute to the presence of polluted dust in urban areas near ports. The neighborhoods of Los Mateos and Santa Lucía, sampled specifically because of their proximity to an industrial area, are apparently not a hotspot of contamination, as their values are in the general trend. The northern part of the city is the least polluted, and it is a newer area with larger avenues and smoother traffic flow.

Once the spatial distribution of pollution and the most dangerous hotspots are known, this work will be communicated to the local competent authorities so that they can be informed and take pollution control measures.

Finally, it would be advisable to carry out a more exhaustive investigation in an area where practically all the maximum values of metals have coincided, especially because it is a tourist and leisure area, and therefore may pose a risk to people's health. Frequent washing of streets is highly recommended to avoid accumulation of dirt in the streets.

**Author Contributions:** P.M.-S., M.J.D.-I. and A.S.-N. conceived and designed this study. P.M.-S., M.J.D.-I., A.S.-N. and A.F.S.-S. were responsible for the data collection, processing and analysis. P.M.-S., M.J.D.-I. and A.F.S.-S. wrote the paper. All authors have read and agreed to the published version of the manuscript.

**Funding:** This research received no external funding.

**Institutional Review Board Statement:** Not applicable.

**Data Availability Statement:** Data are contained within the article.

**Conflicts of Interest:** The authors declare no conflicts of interest.

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
