# Peer review of "Spatial Identification and Hotspots of Ecological Risk from Heavy Metals in Urban Dust in the City of Cartagena, SE Spain"

_sustainability, doi:10.3390/su16010307_

Round 1

Reviewer 1 Report

Comments and Suggestions for Authors

This article investigates the spatial identification and hotspots of ecological risks of heavy metals in dust in Cartagena, Spain, which has certain scientific significance.  I suggest publishing this article after minor revisions.

1, There are too many Figures in this article. It is recommended to integrate them into similar Figures.

2, The author should provide data on the reliability of the testing method, and the instrument parameters of ICP-MS should also be provided.

3, This article states that 11 elements were tested, why only 6 elements were given results?

Author Response

Reviewer 1

This article investigates the spatial identification and hotspots of ecological risks of heavy metals in dust in Cartagena, Spain, which has certain scientific significance.  I suggest publishing this article after minor revisions.

We sincerely appreciate the reviewer's comments, which we have addressed as much as possible.

1, There are too many Figures in this article. It is recommended to integrate them into similar Figures.

Efforts have already been made to integrate them. We think they are all interesting. Figures 2, 4, 6, 8, 10 and 12 for example are already compositions of four distribution maps each.

We have tried to unify some figures, but their information is not clearly distinguishable, so we have decided not to unify them.

2, The author should provide data on the reliability of the testing method, and the instrument parameters of ICP-MS should also be provided.

All the chemicals used were of the highest purity available. High-quality water, obtained using a Milli-Q system (Milli-pore, Bedford, MA, USA), was used exclusively. Standard solutions (1000 µg ml-1) of cupper, lead, nickel, zinc, cadmium and chromium were purchased from Panreac (Barcelona, Spain) and diluted as necessary to obtain working standards. All elements were determined simultaneously with a multi-elemental pattern.

The model has been added Agilent 7900.

Has been added to the methodology section.

3, This article states that 11 elements were tested, why only 6 elements were given results?

Indeed the reviewer is right, we started by measuring all the elements in urban dust but then we focused on only 6 in the work (Zn, Cu, Pb, Cr, Ni and Cd), those we considered most interesting in terms of the ecological risk of contamination. To avoid confusion and errors, we have eliminated the remaining elements.

The remaining elements, which are not studied later in the paper, have been removed in line 132 of section 2.3 of Sample analysis, in table 1 the columns of these elements have been removed, and in line 216 these elements have been removed.

In order to improve the English language skills, the paper will be sent to the translation service of the journal. We will use your English Language Editing Services. We will send the revised manuscript.

Reviewer 2 Report

Comments and Suggestions for Authors

Comments to authors

1.       The significant figures should be taken into account, especially when presenting the standard deviation and the metal concentration. The authors should therefore ensure that the significant figures are corrected in Table 1.

2.       What are the maximum permitted values that the authors compare their results to? It would be important for the readers to see the comparison with the maximum permitted values of metals in the environment.

3.       Instead of Chrome in 3.5, change it to chromium.

4.       What is the relationship between pH and the concentration of the metals observed?

5.       Information on how the samples were collected and prepared is not clear. The authors should give a clear detail on this.

6.       11 metals were studied, but only 6 were compared in Table 2, why? The authors should account for this.

7.       The impact of the metals on the population, especially those in higher concentrations, should be discussed.

8.       What kind of ICP instrument was used and which brand is it? Which standards solutions were used? Purchased from where? What were the concentrations of the standard solutions prepared? Which water was used to prepare the standard solutions? Were the metals analysed simultaneously? Any matrix effects that made the analysis challenging? Was there any sample preparation technique required prior the analysis? These questions should be answered by providing the missing information in the main text.

9.       Sentence construction and grammar needs to be improved in the main text.

Comments on the Quality of English Language

The manuscript require moderate improvement on the quality of English, particularly on sentence construction and to some extent, fixing the grammatical errors.

Author Response

Reviewer 2

We sincerely appreciate the reviewer's comments, which we have addressed as much as possible.

  1. The significant figures should be taken into account, especially when presenting the standard deviation and the metal concentration. The authors should therefore ensure that the significant figures are corrected in Table 1.

A second decimal place has been added to the standard deviation so that it has one more figure than the original data, the concentrations, and has three significant figures in table 1.

  1. What are the maximum permitted values that the authors compare their results to? It would be important for the readers to see the comparison with the maximum permitted values of metals in the environment.

There are reference levels in soils, but not in urban dust, which is an important information gap because urban dust is not soil. In terms of permitted levels in soils, each country, even each region has its own legislation and values and it depends very much on the intended use, a use for a playground, residential or agricultural use is much more restrictive than an industrial use, for example.

As there are no established reference levels in urban dust we only compare with other studies.

In addition, the environmental pollution indexes applied have their own scale that allows us to know the degree of pollution.

  1. Instead of Chrome in 3.5, change it to chromium.

      Thanks. Changed

  1. What is the relationship between pH and the concentration of the metals observed?

      The pH of the urban dust samples has not been determined, as it is insufficient and scarce, but the pH of the surrounding soils is known and they are limestone soils with a neutral or slightly alkaline pH.

      We find this observation very interesting. At these neutral or slightly alkaline pHs, typical of our environment, heavy metals are mostly not bioavailable and are therefore less toxic and dangerous.

  1. Information on how the samples were collected and prepared is not clear. The authors should give a clear detail on this.

      The design that was carried out was systematic, samples were collected every 150 m, obtaining a total of 88 urban dust samples (Figure 1). Sampling was carried out in March 2023 during a period of low rainfall. Most of the samples were taken on roads or parking lots, places close to vehicle traffic, although samples were also collected on sidewalks and in parks. Samples were collected over a 1m2 area, dust was swept over the entire surface using brushes. The collected dust was sieved with a 1 mm sieve (non-metallic) and finally stored in small plastic containers. The samples were ground with an agate mortar and passed through a 50 µm sieve prior to acid digestion with aqua regia.

Attached are some photos of the sampling, in case you need clarification.

  1. 11 metals were studied, but only 6 were compared in Table 2, why? The authors should account for this.

Indeed the reviewer is right, we started by measuring all the elements in urban dust but then we focused on only 6 in the work (Zn, Cu, Pb, Cr, Ni and Cd), those we considered most interesting in terms of the ecological risk of contamination. To avoid confusion and errors, we have eliminated the remaining elements.

The remaining elements, which are not studied later in the paper, have been removed in line 132 of section 2.3 of Sample analysis, in table 1 the columns of these elements have been removed, and in line 216 of the table the columns of these elements have been removed.

  1. The impact of the metals on the population, especially those in higher concentrations, should be discussed.

      We do not have specific health studies on the population of Cartagena, although it would be very interesting. These data are confidential and therefore difficult to obtain. We do know from published studies the harmful effects on human health of each of the metals studied at certain levels and there are health indices, for children and adults, that allow us to assess the possible risks to the population and that we plan to study in a future publication.

  1. What kind of ICP instrument was used and which brand is it? Which standards solutions were used? Purchased from where? What were the concentrations of the standard solutions prepared? Which water was used to prepare the standard solutions? Were the metals analysed simultaneously? Any matrix effects that made the analysis challenging? Was there any sample preparation technique required prior the analysis? These questions should be answered by providing the missing information in the main text.

The model has been added Agilent 7900.

All the chemicals used were of the highest purity available. High-quality water, obtained using a Milli-Q system (Milli-pore, Bedford, MA, USA), was used exclusively. Standard solutions (1000 µg ml-1) of cupper, lead, nickel, zinc, cadmium and chromium were purchased from Panreac (Barcelona, Spain) and diluted as necessary to obtain working standards. All elements were determined simultaneously with a multi-elemental pattern.

Has been added to the methodology section.

  1. Sentence construction and grammar needs to be improved in the main text.

Comments on the Quality of English Language

The manuscript require moderate improvement on the quality of English, particularly on sentence construction and to some extent, fixing the grammatical errors.

In order to improve the English language skills, the paper will be sent to the translation service of the journal. We will use your English Language Editing Services. We will send the revised manuscript.

Reviewer 3 Report

Comments and Suggestions for Authors

Dear authors,

Lines 82-92 should be emphasized because they are listed as partial goals, and as they are listed in the text they may be unrecognized.

Units of measure mg/kg in the entire text should be shown as e.g. 47.7 mg kg-1.

The graphic representation is obtained by interpolation of values, so the question is whether it is better to display it next to each point in the range of obtained values, because by presenting it in this way it is not possible to determine whether the pollution is only at the test point and extends to the environment that may not be contaminated. In the conclusion, it was stated that there was no data on traffic density, but it was carried out approximately by a noise study, which should be explained when presenting the methodology.

In the conclusion, what has been achieved and what measures will be implemented based on the conclusions should be elaborated.

Author Response

Reviewer 3

We sincerely appreciate the reviewer's comments, which we have addressed as much as possible.

Dear authors,

Lines 82-92 should be emphasized because they are listed as partial goals, and as they are listed in the text they may be unrecognized.

They are indeed right. They have been highlighted by adding a dash in front of each of the stated objectives. They have also been separated with a return from the previous paragraph.

Units of measure mg/kg in the entire text should be shown as e.g. 47.7 mg kg-1.

Thank you for your comment. I fully agree, the whole text has been changed mg/kg per mg kg-1.

The graphic representation is obtained by interpolation of values, so the question is whether it is better to display it next to each point in the range of obtained values, because by presenting it in this way it is not possible to determine whether the pollution is only at the test point and extends to the environment that may not be contaminated. In the conclusion, it was stated that there was no data on traffic density, but it was carried out approximately by a noise study, which should be explained when presenting the methodology.

Our objective was to obtain a spatial distribution and to make the maps, the sampling was systematic, sampling every 150 m to make a detailed mapping and we believe that interpolation is the only way to make the maps and to be able to detect hot spots. Now, we fully agree with the reviewer that this interpolation carries the implicit error of assigning values to points and areas that have not been sampled. Only sampled points have an accurate value.

In the methodology of the work (2.3 Sample analysis, lines 127-130), we textually state: "In the absence of data on traffic intensity in the urban area of the city of Cartagena, a study relating noise to traffic was used (Ochoa and Gil de Pareja, 2018), where streets with more noise were considered to be of high traffic intensity, those with less noise were considered to be of medium intensity and, finally, those that did not appear were considered to be of light traffic intensity.”

In the conclusion, what has been achieved and what measures will be implemented based on the conclusions should be elaborated.

Once the spatial distribution of pollution and the most dangerous hot spots are known, this work will be communicated to the local competent authorities so that they can be informed and take pollution control measures. This conclusion has been added.

In order to improve the English language skills, the paper will be sent to the translation service of the journal. We will use your English Language Editing Services. We will send the revised manuscript.
